# A New in Silico Antibody Similarity Measure Both Identifies Large Sets of Epitope Binders with Distinct CDRs and Accurately Predicts Off-Target Reactivity

**DOI:** 10.3390/ijms23179765

**Published:** 2022-08-28

**Authors:** Astrid Musnier, Thomas Bourquard, Amandine Vallet, Laetitia Mathias, Gilles Bruneau, Mohammed Akli Ayoub, Ophélie Travert, Yannick Corde, Nathalie Gallay, Thomas Boulo, Sandra Cortes, Hervé Watier, Pascale Crépieux, Eric Reiter, Anne Poupon

**Affiliations:** 1MAbSilico, 1 Impasse du Palais, 37000 Tours, France; 2Physiologie de la Reproduction et des Comportements, INRAE UMR-0085, CNRS UMR-7247, Université de Tours, 37380 Nouzilly, France; 3Synthélis, BIOPOLIS, 5 Avenue du Grand Sablon, 38700 La Tronche, France; 4Centre Hospitalier Régional Universitaire de Tours, Université de Tours, EA 7501, 37032 Tours, France; 5Inria, Inria Saclay-Île-de-France, 91120 Palaiseau, France

**Keywords:** therapeutic antibody, poly-specificity, off-target, antibody repurposing, in silico method

## Abstract

Developing a therapeutic antibody is a long, tedious, and expensive process. Many obstacles need to be overcome, such as biophysical properties (issues of solubility, stability, weak production yields, etc.), as well as cross-reactivity and subsequent toxicity, which are major issues. No in silico method exists today to solve such issues. We hypothesized that if we were able to properly measure the similarity between the CDRs of antibodies (Ab) by considering not only their evolutionary proximity (sequence identity) but also their structural features, we would be able to identify families of Ab recognizing similar epitopes. As a consequence, Ab within the family would share the property to recognize their targets, which would allow (i) to identify off-targets and forecast the cross-reactions, and (ii) to identify new Ab specific for a given target. Testing our method on 238D2, an antagonistic anti-CXCR4 nanobody, we were able to find new nanobodies against CXCR4 and to identify influenza hemagglutinin as an off-target of 238D2.

## 1. Introduction

Antibodies (Abs) are both key players of the immune system and remarkable tools for research, diagnosis, and therapy. Recombinant monoclonal Abs have been used for 30 years as innovative drugs, and this undeniable success is far from dying out. They possess the advantages of being much more specific for their targets than small chemical compounds, exhibiting better solubility and stability, and presenting lower risks of degradation into toxic metabolites, limiting side effects when compared to chemical drugs. Ab characterization is craggy, and numerous pitfalls have to be avoided.

Off-targets and antigen cross-reactivity have long been an understudied aspect of therapeutic antibody development. This is partly due to the strong belief that antibodies are mono-specific and that, at worst, antibodies could only bind to close homologs of their targets. The other reason is the lack of fast, low-cost, and reliable methods to identify off-targets. Cross-reactivity is still empirically investigated. Immunohistochemical tissue cross-reactivity assays (TCR) constitute the gold standard, being a regulatory requirement before starting first-in-man trials. Off-targets can also be investigated by measuring the binding onto collections of cells overexpressing human membrane proteins [1]. These methods are, unfortunately, neither precise nor exhaustive. Moreover, they are long and costly. Today, no in silico predictive method exists to identify the off-targets of an Ab and to anticipate its cross-reactivity at the early stages of Ab development.

There is increasing evidence that off-targets can be responsible for failure during clinical development [2,3,4]. Consequently, risks associated with cross-reactions should be estimated as early as possible. There is also evidence that existing off-targets can lead to repurposing an existing antibody. This is the case with rituximab, a well-known anti-CD20 antibody used in the treatment of certain lymphoma, leukemia, and allograft rejection. After having observed that two patients treated with rituximab for post-transplant lymphoproliferative disorders experienced complete remission of their concomitant focal segmental glomerulosclerosis (FSGS), the possibility of an off-target effect has been investigated. It was then proven that rituximab binds to sphingomyelin-phosphodiesterase-acid-like-3b (SMPDL-3b), acting as a direct modulator of podocyte function, leading to remission of the nephrotic syndrome. Thus, cross-reactivity is not necessarily bad news and can lead to proposing new applications for existing (or in development) Abs.

The second aspect of Ab development related to the present work is the very high attrition rate [5]. Among the many causes of failure despite promising biological activities, it is not rare that Abs have to be abandoned because of unsuitable biophysical properties (issues of solubility, stability, weak production yields, etc.) [6]. The classical path to solve the problem is to introduce mutations while hoping that the biological properties of the initial Ab will be conserved.

We aim to develop in silico tools to help and accelerate Ab discovery and characterization, starting as soon as the sequence is known. In that scope, we previously published MAbTope, a docking-based method to determine the antibody/antigen interface [7,8]. The success of MAbTope at predicting the epitope of an antibody only starting from the sequence of its variable domains clearly shows that the structural and chemicophysical features of the CDRs are sufficient to define the epitope. We thus hypothesized that two Abs that would share common structural features of their paratopes would recognize the same epitope. As a consequence, the recovered Ab would recognize the target of the seed Ab and, reciprocally, the seed Ab would recognize the target of the recovered Ab. In the first case, it would allow us to identify new binders for a given epitope of a target, having distinct CDRs. In the second, it would identify new targets for the seed Ab, i.e., off-targets. Here we describe a novel in silico method to measure Ab similarity, which relies on a distance measurement between the CDR. Starting from the sequence of a seed Ab, our method allows measuring the similarity with all the Abs in a database. A case study is presented showing the identification of new anti-CXCR4 nanobodies (Nbs) exhibiting similar pharmacological properties to 238D2 and the finding of a new target for 238D2: the influenza virus hemagglutinin.

## 2. Results

### 2.1. Antibody Similarity Measurement

Our initial postulate was that two Abs binding the same target at the same epitope should have common features on their paratopes and that it should be possible to design a similarity measure. It is known that Ab-Ag recognition depends both on structural and physicochemical complementarity and that sequence similarity is not sufficient to predict the fact that two antibodies could have the same epitope specificity. Consequently, we consider both the sequence and the secondary structure of the regions forming the paratope.

For all antibody fragments, the first step consisted in extracting the CDRs using Chothia’s definition [9]. The amino acids were then reassigned into 6 categories according to their physicochemical properties: small (A, G, S, T, C, and P), aromatic (Y, F, and W), hydrophobic (I, L, F, M, and V), polar (N and Q), positively charged (H, K, and R) and negatively charged (D and E). Within each group, the chemical environment created was considered similar enough for our purpose. The secondary structural motifs were annotated as ß-ladder, π-helix, bend, etc., according to Kabsch’s and Sander’s dictionary of protein secondary structures (DSSP) [10]. The similarity between two such encoded Abs was then computed using the similarity measure for sequences of itemsets described by Egho et al. [11]. Consequently, the method gives equal weight to the sequence and secondary structure.

To evaluate our ability to cluster similar paratopes, we built a test set of 987 Ab-target complexes for which the 3D structures are available, corresponding to 447 different protein targets. Two Abs within the set (a pair of Abs), binding to the same target, were considered as competing if the overlap between their epitopes was not null. The resulting test set contains 2807 competing pairs and 1833 non-competing pairs. Pairs of Abs binding to different targets are not considered in this first analysis.

For each pair, we computed the similarity between the two Abs, the sequence identity between the variable domains, and the sequence identity between the 6 CDRs. We determined the thresholds corresponding to three different precision (the ratio of pairs identified as competing that are actually competing) values. The threshold corresponds to the value of the score (or the sequence identity) above which an Ab pair is predicted to compete. We then computed the recall (the ratio of competing pairs that are identified) for the three different fixed precision values. For example, when considering the similarity score, all pairs having a score higher than 73 are predicted as competing. This threshold corresponds to a precision of 1 (all pairs predicted as competing are indeed competing) and a recall of 0.05 (5% of the competing pairs are predicted as competing). The results (Table 1) show that at each of the three precision levels chosen, the similarity leads to a better recall than the two other methods.

### 2.2. Selection of Antibody Fragments Similar to 238D2, An Anti-CXCR4 Nb

The results above demonstrate that our similarity measure is a powerful tool to identify competing pairs among Abs binding to the same target. The next question was to know whether two Abs known to bind to different targets but having high similarity would also compete on each other’s target. This question cannot be addressed using a benchmark since the number of Abs with demonstrated off-targets is limited. Moreover, in those few cases, the antigens on the off-target are not known.

Approximately 30% of the drugs commercialized today are directed against G protein-coupled receptors (GPCRs), making them the largest class of protein targeted in the pharmacopoeia [12]. For decades, the strategy to target a GPCR consisted of high throughput screening of small molecule libraries. However, only a few new synthetic compounds have been approved for therapeutic use during the past decade, showing that this strategy is facing serious limitations. In the meantime, investigations on Ab-based approaches to target GPCRs have erupted. The CXCR4 chemokine receptor is implicated in hematopoietic stem cell dissemination, angiogenesis, HIV-1 entry, and metastatic spread, among other roles [13,14]. CXCR4 is activated by the SDF-1/CXCL12 chemokine. Two anti-CXCR4 single chain Ab fragments have been described, namely the 238D2 and 238D4 nanobodies (Nbs), that antagonize CXCR4/SDF-1-mediated signaling and HIV-1 entry in CD4+ cells (patent number: US 2011/0117113 A1) [15].

The 238D2 Nb is an antagonist of CXCR4 [15], whose 3D structure is not known. We first constructed a homology model of 238D2, using the PDB structure 5NBD as a template, as it was the closest variable domain with a known 3D structure. We then predicted its epitope using MAbTope [8]. The results were consistent with the epitope mapping previously published [15], providing confidence to our 3D model.

We then computed the similarity between 238D2 and all heavy variable domains present in the PDB. We found no variable domain having a score higher than 73, but 11 variable domains had scores higher than 55 (Table 2), thus having a high probability of binding CXCR4.

For the following experimental validations, we only produced the VH domains, ignoring the VL of the described Fab. These Nbs (Nb will also be used for isolated VH domains) were produced in a cell-free system. All Nbs were well produced, except for 4KML and 1SEQ, which had to be abandoned. For the remaining 9 Nbs, the binding to CXCR4 was assessed using flow cytometry on cells transiently overexpressing CXCR4. For 7 out of the 9 antibodies, the bimodal aspect of the fluorescence histograms (Figure 1) demonstrates their binding to the fraction of cells expressing human CXCR4. 2Q76 and 3NH7 were not binding CXCR4 in our experimental setting.

The linear sequences of all 9 Nbs were aligned with 238D2 and an anti-ovalbumin Nb (used as a control in later experiments) using Clustal Omega (EMBL-EBI website). As reported in Figure A1, the CDRs were poorly similar to each other. CDR3, which is known to bear most of the antigen specificity [16], was even the most different of all the CDRs, with a maximal identity to 238D2 of 33%. Because of this poor similarity, none of these VHs would have been anticipated as a suitable candidate for competing with 238D2 when considering only their primary sequences. Indeed, when running a blast with the 238D2 sequence against the whole PDB, none of the 9 candidates appear in the top 100 results.

For further studies, we focused on two of these Ab fragments: 3SM5 and 4N1H, which have the best similarity scores (Table 1) of the VH and VHH, respectively. 3SM5 is the PBD reference number of a complex between the Fab of CH65, a neutralizing Ab, and the receptor binding domain of hemagglutinin from the influenza A virus (A/SOLOMON ISLANDS/3/2006 strain) [17]. 4N1H is the crystallographic structure of the cAB-F11N VHH with its target, the Bacillus licheniformis β-lactamase (Pain C et al., unpublished). Another reason for choosing 3SM5 and 4N1H is that their target proteins that are evolutionarily distant from CXCR4: hemagglutinin (3SM5) and ß-lactamase (4N1H), ensuring that neither their cognate targets nor close homologs are present in the HEK293 cells used in the experiments. We compare the 3D structures of the complexes between the three Nbs and their cognate antigens. Neither the 3D structures of their 3 CDRs nor that of their cognate epitope did look similar (Figure 2). This shows that taking into account both the sequence and the secondary structures deeply modifies the notion of similarity.

### 2.3. Biological Evidence for the Cross-Reactivity of Both 3SM5 and 4N1H on CXCR4 Function

The following question, after the observation of binding, is to investigate a biological effect in live cells. 3SM5, 4N1H, and 238D2 were produced in a bacterial system and were assessed by cytometry for their binding on CXCR4-overexpressing HEK293FT. The three Nbs induced a strong shift in the APC+ cell population (Figure A2).

As previously reported, 238D2 competes with SDF1 for its binding pocket on CXCR4 [15]. Our predictive model of 3SM5 and 4N1H docking on CXCR4 suggested that they share the same epitope as 238D2 (Figure A3). We thus investigated whether 3SM5 and 4N1H also competed with the ligand, as 238D2 does. The HTRF ratio showing the proximity between the receptor and its ligand was measured between a SNAP-tagged CXCR4 (coupled to the Tb donor) and fluorescent SDF1 before competitive experiments in the presence of the Nb (black dot, Figure 3A). The dose-dependent decrease in the energy transfer between CXCR4 and SDF1 showed that the 3 Nbs displaced SDF1 from its receptor and hence that SDF1, 238D2, 3SM5, and 4N1H share the same binding site on CXCR4. The non-relevant anti-ovalbumin Nb did not displace SDF1. The IC50 was very similar for the 3 Nb (58.4, 69.1, and 39.8 nM for 238D2, 3SM5, and 4N1H, respectively). However, the pharmacological profiles of the 3 Nb may be slightly different since the displacement was incomplete with 3SM5 and 4N1H when compared to 238D2.

CXCR4 activates G proteins and recruits β-arrestins on its intracellular region upon SDF1 binding [18]. 238D2 has been characterized as an antagonist of CXCR4 according to different Gi-protein-dependent read-outs, such as IP1 production or cAMP-responsive element (CRE) activation [15]. We sought to investigate to which extent 3SM5 and 4N1H binding to CXCR4 altered Gi protein- and β-arrestin-dependent signaling pathways. First, HEK293FT cells were co-transfected with the receptor and the CAMYEL cAMP-sensitive BRET sensor. We ran the experiment in the presence of the ligand, and since CXCR4 couples to Gi protein, basal cAMP production was induced by forskolin. Increasing concentrations of 238D2, 3SM5, or 4N1H, but not the anti-ovalbumin Nb, reversed SDF1-induced coupling to Gi, showing that they all behave as antagonists (Figure 3B).

We also studied the effect of the Nb on SDF1-mediated β-arrestin recruitment to CXCR4 that had not been examined previously. HEK293FT cells were co-transfected with a CXCR4-RLuc fusion construct and YPET-tagged β-arrestin 2. In response to SDF1, the BRET signal between the 2 constructs increased, showing that β-arrestin 2 was actually recruited to CXCR4 (Figure 3C). Increasing doses of 238D2, 3SM5, and 4N1H, but not control anti-ovalbumin Nb, dampened this recruitment, consistent with antagonistic activity. We performed the same β-arrestin assay on 2 other GPCRs to ensure the specificity of the three Nb for CXCR4 (Figure A4). Neither 238D2 nor 3SM5 or 4N1H inhibited β-arrestin 2 recruitment to the vasopressin 2 receptor (Figure A4A) or to the luteinizing hormone receptor (Figure A4B).

These data demonstrated that our database-recovered Nbs mimic 238D2 in all pharmacological aspects studied (full cross-functionality) and could therefore be used as surrogates of 238D2 to antagonize CXCR4.

### 2.4. H1N1 Identified As a Target of 238D2

3SM5 and 4N1H bind to the 238D2 target, the CXCR4 receptor. According to our postulate, 238D2 is expected to bind to both the targets of 3SM5 and 4N1H. We focused on the 3SM5 antigen: the influenza A hemagglutinin H1 from the A/SOLOMON ISLANDS/3/2006 strain, which was commercially available as a fusion protein with the neuraminidase N1 (H1N1). By ELISA, we first showed that 238D2 actually binds to H1N1, as 3SM5 does (Figure 4A). 4N1H did not interact with hemagglutinin significantly, which may be explained by its higher distance score to 3SM5 (54, 86). The direct interaction between 238D2 and H1N1 was confirmed by immuno-precipitating H1N1 on beads solely in the presence of the Nb (Figure 4B). In addition, the affinity of 238D2 for H1N1 was measured by biolayer interferometry (Figure 4C). Streptavidin-coupled biosensors were loaded with the biotinylated AviTag-238D2, and the association to H1N1 was evaluated at several concentrations from 87.5 to 875 nM. The K_D_ was evaluated as 119 ± 1.78 nM. Using the same procedure, the K_D_ between AviTag-3SM5 and H1N1 was in a similar range (236 ± 13.7 nM) (Figure 4C).

These results show that the query fragment 238D2 binds to the targets of similar Ab fragments, and hence that our method is potentially efficient for identifying off-targets.

## 3. Discussion

The method described here represents a new way of quantifying the similarity between the CDRs of different Abs. Homology between proteins is classically defined as identity or homology between their linear sequences, reflecting their evolutionary proximity. However, linear sequence proximity cannot be by itself a predicator of common epitope recognition. The recognition of one Ag by an Ab relies on both chemical and geometrical complementarity, which allows bringing the partners close enough so that non-covalent bonds can be formed. Hence, in the method presently described, we simultaneously considered the amino acid properties and the local 3D structures they belong to. As a consequence, our method succeeded in clustering Abs not only on their evolutionary proximity but also on their chemicophysical and spatial properties, conferring their specificity for their cognate epitope.

The data (Ab sequences and described off-targets) used for this study were extracted from the PDB. For the substitution task, which consists in finding a new antibody that binds to the target of the query antibody, the database can be greatly increased by the addition of NGS data that can be found in the NCBI Sequence Read Archive (SRA). However, to detect an off-target for a query antibody, the major limitation of our method is that we need an antibody binding this off-target at the right epitope in our database. The performance of the off-target detection task hence depends on the number and diversity of antibodies in the database used for searching. The PDB, restricted to Abs (about 6000 variable domains), is clearly not sufficient to exhaustively predict off-targets and to evaluate the safety of an Ab. We are continuously extending our own databases by collecting information from filed patents to increase our collection of Ab sequences with known targets. Our database of antibodies having known cognate targets now contains more than 50,000 non-redundant antibodies, covering more than 5000 targets. Among these targets, about 1500 are human proteins, and another 1000 correspond to non-human mammalian proteins.

Abs are undeniably much more specific for their targets than small chemical compounds. However, much evidence shows that Abs do have off-targets. The notions of Ab specificity and Ab cross-reactivity were enunciated in the 1950s, when the variety of the Ig repertoire started to be grasped and opposed to the variety of the antigen determinants in nature [19]. In the late 1980s, it was evaluated that up to 3.5% of 600 anti-viral monoclonal antibodies tested cross-reacted in normal mice tissues [20]. At that time, molecular mimicry displayed by the recognized antigens offered a satisfying explanation, but since then, it has been demonstrated that Ab also undergoes induced fit, which consists of structural rearrangements between bound and unbound states, which depend on the facing antigen [21,22,23]. This flexibility seems to be mainly located in the CDR H3 loop, the most implicated in Ag recognition. Therefore, an Ab is able to recognize different epitopes not only because they share common structural features but also thanks to its own capacity to adapt to different epitope structures.

Rationalizing Ab cross-reaction has greatly helped to understand the etiology of allergies and autoimmune diseases [24,25]. Pathogen-originated autoimmune diseases are suitable illustrations of what cross-reactivity can induce, such as Lyme arthritis after *Borrelia burgdorfei* infections [26], Guillain-Barré neurologic syndrome after *Campylobacter jejuni* infection [27], or Chagas heart disease, which is a complication in as much as 30% of the patients infected with the parasite *Trypanosoma cruzi* [28] (see [24] for a review). In our case study, we identified several Ab similar to 238D2, which covered a diverse panel of secondary targets. Direct proof was established for the targeting of hemagglutinin from the H1N1 influenza virus (A/SOLOMON ISLANDS/3/2006 strain), which is, therefore, an off-target. This would have no serious consequence if the Ab were used in therapy since it is not a human protein. However, since Ab 3SM5 does bind CXCR4, the question of sensitivity created by the influenza virus and an autoimmune reaction against CXCR4 could be raised.

We also demonstrated that the VH domains of the Ab referenced as 3FFD, 3L95, 4HCR, and 4KI5 in the PDB bound the CXCR4. These Ab were originally raised against human proteins: the parathyroid hormone-related protein (PTHrP) [25], Notch1 [29], the mucosal vascular addressin cell adhesion molecule 1 (MAdCAM-1) (Springer et al., not published), and the human coagulation factor VIII [30], respectively. PTHrP and Notch have been both related to cancer: PTHrP for inducing bone metastasis from breast cancer [31] and Notch for being implicated in abnormal cell growth and fate [32]. 3FFD and 3L95 are inhibitory Abs that have anti-cancerous ambitions. 3FFD impedes PTHrP binding to its receptor [33], and 3L95 protects against proteases Notch negative regulatory region (NRR) and avoids the activation of Notch signaling [29]. The inhibition of PTHrP and Notch signaling by the use of 238D2 could potentially be an issue in the case of therapeutic application.

4HCR is the crystal structure of PF-00547659 Ab (from Pfizer, New York, NY, USA) in complex with MAdCAM-1, which is an addressin expressed on endothelial cells directing myeloid cells on inflammation sites. PF-00547659 was expected to limit the homing of lymphocytes in the intestinal tract in patients with Crohn’s disease and consequently help tissue repair, but it failed to prove significant efficacy in phase II trials [34]. PF-00547659 may thus be repositioned on a new anti-CXCR4 application.

Finally, 4KI5 is the reference of the structure of 2 Fab (3E6 and G99, only 3E6 displays high similarity with 238D2) complexed with the coagulation factor VIII [30]. 3E6 occludes the residues involved in the binding of factor VIII to both the von Willebrand factor and the platelets membrane phospholipids. If proven, the binding of 238D2 to factor VIII could, in the less severe scenario, alter the pharmacokinetic and the biodistribution of the 238D2 and, in the worst-case scenario, have a negative effect on blood coagulation, similar to what is observed in hemophilic patients. It is, however, important to notice that the final molecule developed by Jähnichen and colleagues is a biparatopic Nb constituted of 238D2 and another anti-CXCR4 Nb, 238D4 [15].

For the past decades, the development of therapeutic Ab has followed an exponential increase. Their pharmacological properties (specificity for the target, extended half-life…) and pharmacological diversity (from target activity modulation to target neutralization) made them revolutionary tools to treat diseases hardly targetable by classical chemistry. Today, more than 140 monoclonal Abs are FDA-approved for therapeutic use (mostly to fight cancers and autoimmune diseases) (https://www.antibodysociety.org/resources/approved-antibodies/, accessed on 12 July 2022), and hundreds are undergoing clinical trials. Considering that, in the past decade, over a thousand Abs per year reached the public domain (https://tabs.craic.com, accessed on 12 July 2022) (conferences, patents, and papers), the calculation can be made that far less than 1% of the Ab reach the pharmacopoeia. This bottleneck is undoubtedly multifactorial, and off-target issues may obviously not be held responsible for the entirety of these failures. Nonetheless, if we consider off-targets as an intrinsic property of the Ab, tools aiming at efficiently and exhaustively identifying the off-targets need to be applied as early as possible in the Ab development process. As already mentioned, the prediction through our method of the binding of a query antibody to a protein different from its cognate target necessitates prior knowledge of antibody sequences having this protein as a cognate target.

However, off-targets are not always detrimental and can sometimes have beneficiary effects, the most emblematic example certainly being rituximab. As already evoked in the introduction, its off-target binding to the sphingomyelin-phosphodiesterase-acid-like-3b (SMPDL-3b) [35] confers efficiency in the treatment of focal segmental glomerulosclerosis (FSGS), including multi-drug-resistant forms [36]. This example clearly demonstrates that off-target detection can also lead to antibody repurposing.

The large amount of data already accumulated about Abs constitutes a huge reservoir of information. By exploiting a very small part of these data, we succeeded in developing a method whose applications are: (i) the early identification of putative off-targets and the hasty forecast of cross-reactions, but also (ii) the identification of large sets of CDRs-distinct of potential binders for the same antigen, and (iii) drug repositioning, i.e., the finding of new targets and functions for an existing Ab. Importantly, the off-target search is very fast (a few seconds for one antibody sequence), can be applied to large batches of antibodies (tens of thousands), and only requires the antibody sequence. Consequently, this method can be applied very early in the discovery process, as soon as sequences of putative binders are known. It could thus become a criterion for choosing the right lead, which will save much time and money later in the process.

## 4. Materials and Methods

### 4.1. MAbCross Itemsets Similarity

For each variable domain sequence (query Ab and Ab of the search database), all CDRs fragments were delineated using Chothia’s numbering [9]. Residues are binned into six categories according to their chemical properties: small (AGSTCP), aromatic (YW), aliphatic (ILFMV), polar (NQ), negatively (DE), and positively (HKR) charged. The secondary structure of each amino acid is made according to Kabsch’s and Sander’s dictionary of protein secondary structures (DSSP). These two pieces of information are aggregated in a single code for each amino acid of the CDRs

The similarity between two encoded sequences is then computed using the method introduced by Egho et al. [11]. Briefly, the methods consist in enumerating all subsequences common to the two compared encodings.

### 4.2. Nanobody Sequences

The Ab that are not bona fide Nb were restricted to their VH domain to generate a pseudo-Nb. The Nb coding sequences were generated (EMBOSS Backstranseq software, EMBL-European Bioinformatics Institute, Cambridge, U.K.) from the protein sequences shown in Table 3.

### 4.3. Bacterial Expression and Purification

The Nb sequence genes were synthesized and cloned in pET28b(+) (Eurofins Genomics, Ebersberg, Germany). Protein expression was induced by 1 mM IPTG addition to the medium of E. coli BL21 DE3 strain for non-biotinylated Nb and of BirA biotin ligase-expressing E. coli for Avi-tagged constructs. In this latter case, the culture medium was complemented with 10 µM biotin (ThermoFischer Scientific, Waltham, MA, USA). 6xHis-directed purification of the Nb was performed in denaturing conditions using Protino^®^ Ni-NDA columns according to the manufacturer’s instructions (Macherey Nagel, Düren, Germany). Nb was dialyzed, renatured, and concentrated using 5 kDa cut-off Vivaspin (Sartorius Stedim GmbH, Goettingen, Germany). Briefly, Nb solutions were concentrated 5 successive times using in 0.1 M Tris HCl pH 8-0.5 M NaCl containing 4 M urea, 2 M urea, 1 M urea + glutathione red/ox (7.5 mM and 0.75 mM, respectively), 0.5 M urea + glutathione red/ox, and finally in Tris HCl containing 5% glycerol. Nb concentrations were measured using a Nanodrop 2000 (ThermoFisher Scientific, Waltham, MA, USA).

### 4.4. Cell-Free Protein Expression and Purification

Nanobody genes were synthesized by ProteoGenix (Schiltigheim, France) and subcloned into a specific vector designed for high-level expression of His-tagged proteins in a cell-free expression system. The vector contains all regulatory elements necessary for in vitro expression based on a combination of T7 RNA polymerase. A cell-free reaction was carried out at 20 °C for 16 h with gentle agitation at 150 rpm by using an *E. coli* extract and energy mix developed by Synthelis SAS (La Tronche, France). Each reagent was carefully stored and handled in an RNase-free environment. The T7-based transcription was obtained using a 15 μg/mL plasmid solution. The cell-free synthesis reaction mixture was centrifuged at 15,000 rpm for 10 min at 4 °C. The supernatant was then recovered and diluted 2 times with 150 mM NaCl, 50 mM Tris, pH 8, 10 mM imidazole, and loaded on MagneHis beads (Promega, Madison, WI, USA) overnight (batch mode). Washes were carried out in the same buffer, and elution was performed in three fractions containing 500 mM imidazole. Protein quantification and quality were assessed by UV absorbance at 280 nm with the Nanodrop 2000 (ThermoFisher Scientific, Waltham, MA, USA) and Western blot analysis using a rabbit anti-His tag antibody.

### 4.5. ELISA

Influenza A H1N1 (A/SOLOMON ISLANDS/3/2006) (Interchim, Montluçon, France) was coated on Maxisorp 96-well plates (2 µg/mL in PBS-1% milk) overnight at 4 °C. After 3 washes in PBS-0.01% Tween-20, the Nbs were added (1 µM in PBS milk) for 1 h. Plates were washed thrice and incubated with HRP-coupled anti-c-myc 9E10 antibody (Abcam, Cambridge, MA, USA) diluted at 1:2000 in PBS milk. After three final washes, the luminescence signal (Pierce ECL Western Blotting Substrate, ThermoFischer Scientific, Waltham, MA, USA) was read in a multiplate reader (Mithras^2^ LB 943 Monochromator Multimode Reader, Berthold Technologies, Thoiry, France).

### 4.6. Immuno-Precipitation and Western Blotting

Five hundred microliters of protein A and protein G-coated magnetic beads (Merck Millipore, Armstadt, Germany) were equilibrated in TNET buffer (20 mM Tris HCl pH 8, 137 mM NaCl, 1% Nonidet P-40, 2 mM EDTA) and loaded with 10 µL of anti-c-myc 9E10 antibody (Abcam, Cambridge, MA, USA) overnight at 4 °C. After 3 washes in TNET, 10 µg of c-myc- and Avi-tagged biotinylated Nb and/or 10 µg of influenza A H1N1 (A/SOLOMON ISLANDS/3/2006) (Interchim, Montluçon, France) were added to the beads in 500 µL TNET. The immune complexes formed under mild rotation for 2 h at 4 °C. The beads were finally washed thrice and suspended in protein buffer (0.02% bromophenol blue, 125 mM Tris HCl pH 6.8, 20% Glycerol, 4% SDS, 10% β-mercapto-ethanol). Protein suspensions were boiled at 100 °C for 5 min, separated by SDS-PAGE, and gels were transferred onto nitrocellulose membranes that were afterward saturated in Tris Buffered Saline—0.01% Tween-20 (TBS-T)—3% BSA for 30 min at room temperature. Primary anti-His antibody (Anti-6X His tag antibody-ChIP Grade, Abcam, Cambridge, MA, USA) diluted 1:2000 in TBS-T-BSA was used to probe the poly-histidine tag located on both the Nb and H1N1. After overnight incubation and 3 washes in TBS-T, the blot was revealed with the IRDy 680 goat anti-rabbit secondary antibody (LI-COR, Lincoln, NE, USA) diluted to 1:10,000 in TBS-T-BSA. The biotinylated Avi-tagged Nb was also probed with a 1:10,000 diluted AF680-coupled streptavidin. Membranes were washed three times and scanned on Odyssey CLx (LI-COR, Lincoln, NE, USA).

### 4.7. Biolayer Interferometry (BLI)

All the measurements were performed with the Octet RED96 System (Pall Forte Bio, Fremont, CA, USA), in Pall kinetics buffer at 30 °C and by shaking at 1000 rpm. Biotinylated AviTag-238D2 (100 nM) was immobilized for 300 s on streptavidin-coated sensors (SA) and left for equilibration for 120 s in a kinetics buffer. Typical capture variability within a row of eight tips did not exceed 0.2 nm. Binding was assessed at 50, 30, 20, 15, 10, 7.5, and 5 µg/mL of influenza A H1N1 (A/SOLOMON ISLANDS/3/2006) for 300 s. Parallel correction was carried out by subtracting the association of H1N1 on immobilized non-relevant biotinylated protein. Data were analyzed using Octet Software version 9.0. Experimental data were fitted with the binding equation describing a 1:1 interaction. Global analyses of the data sets assuming that binding was reversible (full dissociation) were carried out using the nonlinear least-squares fitting, allowing a single set of binding parameters to be obtained simultaneously for all concentrations used in each experiment.

### 4.8. Flow Cytometry

The human CXCR4 cDNA in pcDNA3.1 was purchased from the cDNA Resource Center, University of Missouri-Rolla, USA. HEK293FT cells were transfected with 5 µg of plasmid DNA and 15 µL Metafectene (Biontex Laboratories GmbH, München, Germany) per 10 cm^2^ dish, according to the manufacturer’s instructions. Forty-eight hours after transfection, cells were starved for 45 min in PBS—10 mM Hepes and seeded at 250,000 cells per tube in PBS, 2 mM EDTA, 1% BSA. Nbs (100 nM as final concentration) were incubated with the cells for 2 h at 4 °C. After 2 washes in 2 mL PBS-1% BSA-2 mM EDTA, cells were incubated with allophycocyanin-coupled anti-6xHis antibody (AD1.1.10 clone, Abcam, Cambridge, MA, USA) diluted to 1:100 for 1 h at room temperature. Cells were washed again twice and analyzed with the MACSQuant Analyzer 10 (Miltenyi Biotech GmbH, Bergisch Gladbach, Germany). The data were analyzed with FlowJo (FlowJo LLC, Ashland, OR, USA).

### 4.9. HTRF-Based Displacement Experiments

The buffer (Tag-lite Buffer), the labeling dye (Lumi4-Terbium), the d2-coupled SDF1/CXCL12, SNAP-CXCR4 expression vector were all obtained from Cisbio Bioassays, Codolet, France. HEK293 FT cells were transfected with the SNAP-tagged hCXCR4. After 48 h of expression, cells were labeled with Lumi4-Terbium and seeded at 20,000 cells per well in 384-well plates. Cells were exposed to Nb at concentrations from 1 pM to 10 µM for 1 h before adding 12 nM of d2-coupled SDF1. Plates were read on a Mithras^2^ LB 943 Monochromator Multimode Reader (Berthold Technologies, Thoiry, France).

### 4.10. cAMP Production and β-Arrestin 2 Recruitment Assayed by BRET

The CXCR4-Rluc8, the YPET-β arrestin 2, and the CAMYEL constructions were provided by Dr. Mohammed Akli Ayoub (Al Ain University, United Arab Emirates). HEK293FT cells were transfected in 96-well plates with 50 ng of plasmid DNA and 0.5 µL of Metafectene (Biontex Laboratories GmbH, München, Germany) per well, according to the manufacturer instructions. Forty-eight hours after transfection, Nbs were added to the cells in 10 mM PBS—1 mM Hepes and incubated for 2 h at 37 °C. Recombinant SDF1-α (Peprotech, Rocky Hill, NJ, USA) was added for 15 min at a final concentration of 50 nM. Coelenterazin H (Interchim, Montluçon, France) at a final concentration of 50 µM was added to the medium right before the measurement. Forskolin (Sigma Aldrich-Merck, Merck KGaA, Darmstadt, Germany) at a final concentration 100 µM was mixed to Coelenterazin H in CAMYEL assays. Plates were read on a Mithras^2^ LB 943 Monochromator Multimode Reader (Berthold Technologies, Thoiry, France).

### 4.11. Statistics

Graphs were built and data analyzed using Prism software (GraphPad Software Inc., La Jolla, CA, USA). ANOVA tests were performed, and statistical significance was indicated as letters. Identical letters show no significant difference, whereas different letters show significant differences at *p* > 0.05.

## Figures and Tables

**Figure 1 ijms-23-09765-f001:**
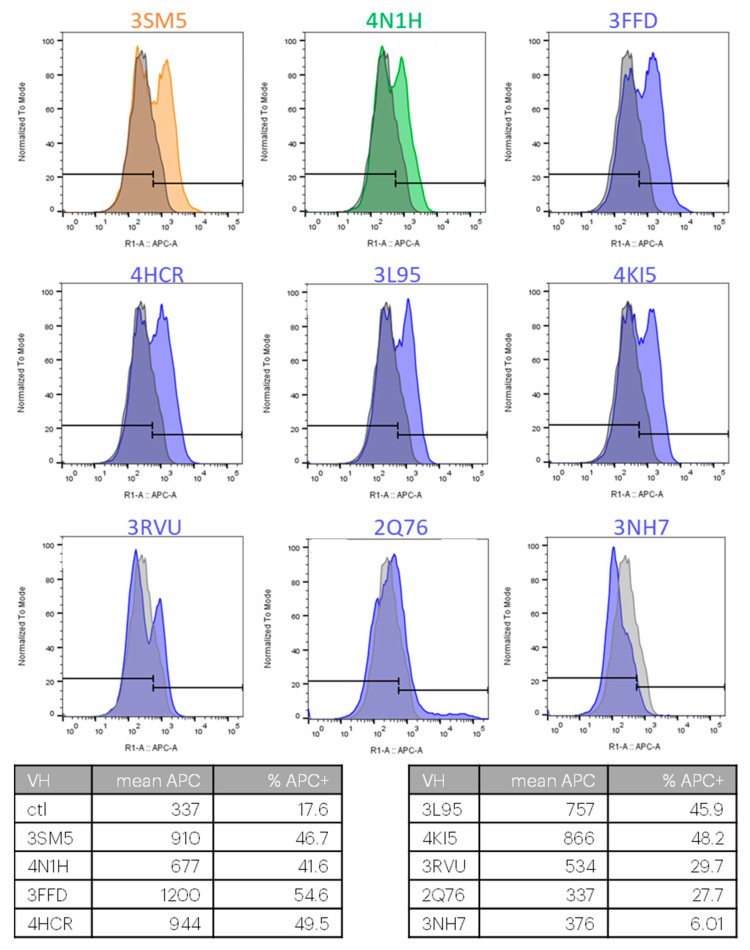
Binding of VH and VHH domains to CXCR4 expressing cells. Nb were produced in a cell-free system. All Nb were incubated with hCXCR4-overexpressing cells and revealed by an APC-conjugated anti-His antibody. The binding of a control Nb is shown in gray.

**Figure 2 ijms-23-09765-f002:**
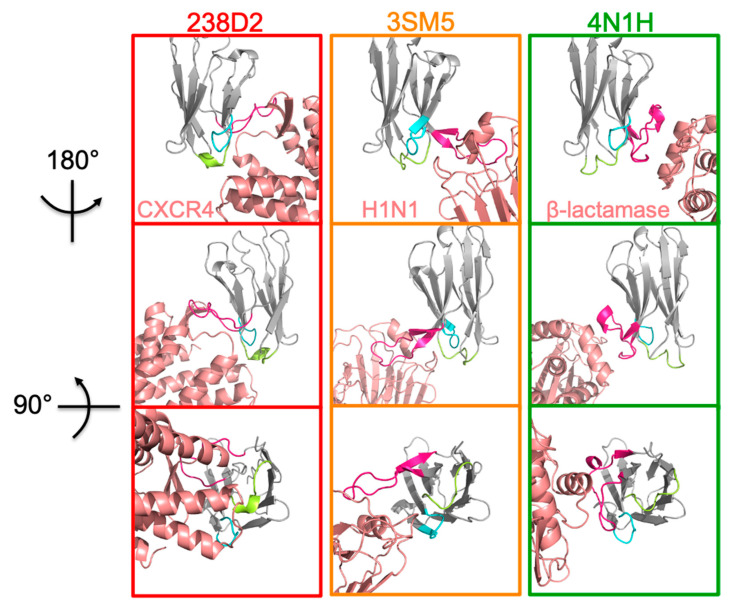
Structural comparisons of 238D2, 3SM5, and 4N1H with their cognate targets. 238D2 on CXCR4 (left panels, MAbTope prediction), 3SM5 on hemagglutinin (middle panels, crystallographic structure), and 4N1H on β-lactamase (right panels, crystallographic structure). The Nb frameworks are shown in gray, CDR1 in blue, CDR2 in green, and CDR3 in magenta. The antigens are shown in salmon.

**Figure 3 ijms-23-09765-f003:**
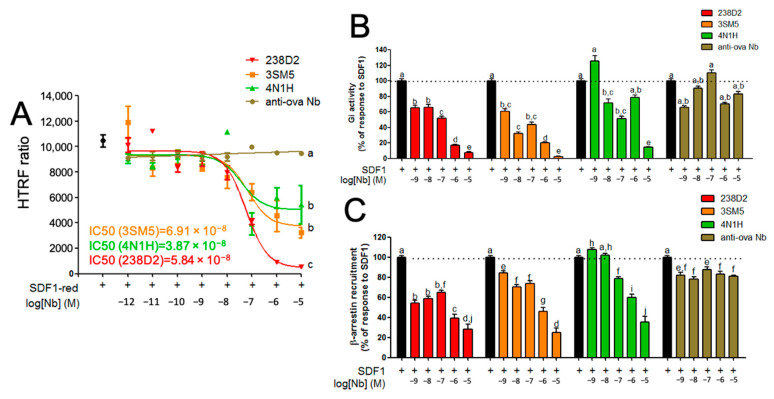
3SM5 and 4N1H bind CXCR4 and trigger the same intracellular signaling as 238D2. (**A**) SDF1 displacement. HEK293FT cells expressing the Terbium-linked SNAP-tagged CXCR4 were incubated with d2-coupled SDF1 to measure the ligand binding as an initial FRET signal (black dot). Increasing doses of the Nbs (238D2 in red, 3SM5 in orange, 4N1H in green, and the anti-ovalbumin Nb in brown) from 1 pM to 10 µM were added. Results are shown as the FRET ratio of emission light of the acceptor (d2) divided by the emission of the donor (Tb). (**B**) Gi protein activity. HEK 293FT cells transiently expressing the hCXCR4 and the CAMYEL cAMP sensor were incubated with forskolin (100 µM) to induce basal cAMP production and SDF1 (50 nM) to induce Gi activation (black bars). The BRET ratio (emission of acceptor/emission of donor) was calculated while increasing Nb concentration from 1 nM to 10 µM. (**C**) β-arrestin recruitment to CXCR4. In HEK293FT cells transiently expressing Rluc8-fused CXCR4 and YPET-β-arr2, β-arrestin 2 recruitment to the CXCR4 was induced with 50 nM SDF1 (black bars). BRET ratio was measured at indicated concentrations of Nb (from 1 nM to 10 µM). Identical letters show no statistical difference, and different letters show statistical differences at *p* < 0.05.

**Figure 4 ijms-23-09765-f004:**
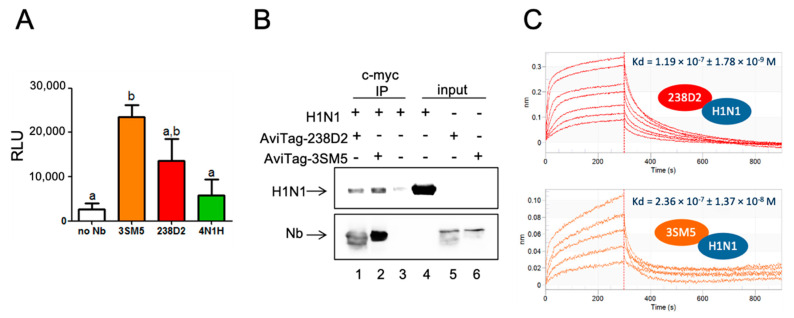
238D2 binds A/SOLOMON ISLANDS/3/2006 hemagglutin. (**A**) ELISA on H1N1. H1N1 was coated on MaxiSorp plates and incubated with the myc-tagged Nb at 1 µM. Nb was revealed with an HRP-coupled anti-c-myc antibody. Here are shown luminescence signals in mean +/− sem obtained from 5 independent experiments. Identical letters show no statistical difference, and different letters show statistical differences at *p* < 0.05. (**B**) Nb-directed immuno-precipitation of H1N1. Magnetic beads were coupled to an anti-c-myc antibody and exposed to a mixture of H1N1 and the biotinylated c-myc-tagged 238D2 (AviTag-238D2, lane 1), or the biotinylated c-myc-tagged 3SM5 (AviTag-3SM5, lane 2) or to H1N1 alone (lane 3). In lanes 4, 5, and 6 are shown the inputs. The beads eluate, and the input samples were analyzed by Western blot. H1N1 was revealed by an anti-His tag primary antibody (upper panel), and the Nb by an AlexaFluor680-coupled streptavidin (lower panel). (**C**) BLI streptavidin sensors were loaded with biotinylated AviTag-238D2 (top) or AviTag-3SM5 (bottom). Association of H1N1 on the 238D2 or 3SM5 was measured at 84.75 (bottom curve), 127.1, 169.5, 254.2, 339, 508.5 and 847.5 nM (upper curve). The dissociation occurred in the running buffer. The K_D_ indicated on the graph was obtained from the simultaneous fit of the 5 curves.

**Table 1 ijms-23-09765-t001:** Recall values corresponding to fixed precision values for similarity score, sequence identity, and CDR identity. Corresponding threshold values are also given.

	**Thresholds**		
	**Precision = 1**	**Precision = 0.95**	**Precision = 0.75**
This work	73	55	34
Sequence identity	0.87	0.82	0.53
CDR identity	0.7	0.65	0.3
	**Recall**		
This work	0.05	0.11	0.7
Sequence identity	0.04	0.08	0.66
CDR identity	0.04	0.05	0.62

**Table 2 ijms-23-09765-t002:** Variable domains of the PDB (version 15 April 2015) similar to nanobody 238D2, with score above 55.

Accession	Format	Score	Target Antigen
3SM5 ^1^	Fab	64.52	Hemagglutinin
3FFD	Fab	59.88	Parathyroid hormone-related protein
4N1H	VHH	59.87	Beta-lactamase
4KML	VHH	57.58	Human prion protein
2Q76	Fab	57.58	Hen egg white lysozyme
4HCR	Fab	56.03	Mucosal addressin cell adhesion molecule 1
3L95	Fab	55.34	Neurogenic locus Notch homolog protein 1
3RVU	Fab	55.34	Der f1 major allergen
3NH7	Fab	55.28	Bone morphogenetic protein receptor type-1A
1SEQ	Fab	55.28	Neurotrophin receptor TrkA
4KI5	Fab	55.27	Coagulation factor VIII

^1^: 3SM5 has been replaced by 5UGY in the PDB.

**Table 3 ijms-23-09765-t003:** Antibody sequences.

Name	Format	Sequence
238D2 ^1^	VHH	EVQLVESGGGLVQTGGSLRLSCAASGFTFSSYAMSWVRQAPGKGLEWVSGIKSSGDSTRYAGSVKGRFTISRDNAKNMLYLQMYSLKPEDTAVYYCAKSRVSRTGLYTYDNRGQGTQVTVSS
3SM5 ^2^	VHH	MEVQLVQSGAEVKKPGASVKVSCKASGYTFTDYHINWVRQAPGQGLEWMGWIHPNSGDTNYAQKFQGWVTMTRDTAISTAYMEVNGLKSDDTAVYYCARGGLEPRSVDYYYYGMDVWGQGTTVTVSS
3FFD ^2^	Fab	EVQLVESGGDLVKPGGSLKLSCAASGFTFSSYGMSWIRQTPDKRLEWVATISSGGSYTYYPDSVKGRFTISRDNAKNTLYLQMSSLKSEDTAMFYCARQTTMTYFAYWGQGTLVTVSS
4N1H ^2^	Fab	MEVQLQESGGGLVQAGASLKLSCAASGRTFSSYAMGWFRQAPGKEREFVAAISRSGGDTKYADSVKGRFAISRDNDKNTVWLRMNSLKPEDTAVYYCAATTYASLSDTYIGEHIYDDWGQGTQVTVSS
4KML ^2^	VHH	AVQLQESGGGLVQPGGSLRLSCAASGRTFSSYNMGWFRQAPGKGREFVASITSSGDKSDYTDSVKGRFTISRDNAKNTMYLQMNNLKPEDTATYYCARGLGIYIIRARGGYDHWGQGTQVTVSS
2Q76 ^2^	Fab	EVQLEQSGAELMKPGASVKISCKATGYTFTTYWIEWIKQRPGHSLEWIGEILPGSDSTYYNEKVKGKVTFTADASSNTAYMQLSSLTSEDSAVYYCARGDGFYVYWGQGTTLTVSS
4HCR ^2^	Fab	QVQLVQSGAEVKKPGASVKVSCKASGYTFTSYGINWVRQAPGQGLEWMGWISVYSGNTNYAQKVQGRVTMTADTSTSTAYMDLRSLRSDDTAVYYCAREGSSSSGDYYYGMDVWGQGTTVTVSS
3L95 ^2^	Fab	EVQLVESGGGLVQPGGSLRLSCAASGFTFSSYWIHWVRQAPGKGLEWVARINPPNRSNQYADSVKGRFTISADTSKNTAYLQMNSLRAEDTAVYYCARGSGFRWVMDYWGQGTLVTVSS
3RVU ^2^	Fab	EVQLQESGPGLVKPSQSLSLTCTVTGYSITSDYAWNWIRQFPGNKLEWMGYISYSGTTSYNPSLKSRISITRDTSKNQFFLQLNSVTTEDTATYYCGRTGVYRYPERAPYWGQGTLVTVSA
3NH7 ^2^	Fab	QVQLVESGGGLVQPGGSLRLSCAASGFTFSNYTLNWVRQAPGKGLEWVSYTSSSGSLTGYADSVKGRFTISRDNSKNTLYLQMNSLRAEDTAVYYCARERWHVRGYFDHWGQGTLVTVSS
1SEQ ^2^	Fab	EVKLVESGGGLVQPGGSLKLSCAASGFTFSTYTMSWARQTPEKKLEWVAYISKGGGSTYYPDTVKGRFTISRDNAKNTLYLQMSSLKSEDTALYYCARGAMFGNDFKYPMDRWGQGTSVTVSS
4KI5 ^2^	Fab	QIQLVQSGPELKKPGKTVKISCKASDYTFTDYSLHWVKQAPGKGLKWMGWINTETGDPAYADDFKGRFAFSLETSVRTAYLQINNLKNEDTAIYFCAREDDGLASWGQGTTLTVSS
Anti-Ova ^3^	VHH	EVQLQESGGSGQAGGSLRLSCAASGDTVRTMAWFRQAPGQEREGVAGFNLPISRPYYADGMKARFTISGDKSKNTVTLQMDNLAPEDTANYYCAATRYTLDLSSRIFQGDFDHWGHGTQVTVSS

^1^: patent US 2011/0117113; ^2^: RCSB; ^3^: internal.

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
