# Peer review of "A New in Silico Antibody Similarity Measure Both Identifies Large Sets of Epitope Binders with Distinct CDRs and Accurately Predicts Off-Target Reactivity"

_ijms, 2022, doi:10.3390/ijms23179765_

Round 1

Reviewer 1 Report

The authors describe a simples and novel way of identifying antibodies that may cross-react with any particular antigen for which an antibody is already available. The data are interesting also in the context of auto-immune diseases. There are however a number of points that need to be clarified and the presentation of the data needs improvement.

Major points:

1.       Figure 1: It would be nice to see CXCR4 expression on transfected HEK293 cells (% and MFI), seen with anti-CXCR4 238D2 nanobody, to be then compared to binding with other VHH or VH molecules. Since CXCR4 transfection is transient, heterogeneous expression of CXCR4 could have been obtained. This should be shown. In general, it would have been nice if authors had used a stably transfected HEK293 cell line expressing CXCR4 on >90% of cells.

2.       Figure 1: In the histograms, it is not necessary to indicate % APC+ and APC- since these data are already shown in the table below. Table should report in the columns mean fluorescence intensity and % positive cells. Median APC is not a clear label.

3.       Figure 1: It would be nice to see also the results of binding of all VH and VHH to control non-transfected HEK293 cells.

4.       Figure 3+4: what do the labels a,b and c mean? in panel A and above bars in panels B and C

5.       Line 231, authors mention Fig.3D which does not exist. Please explain.

6.       Lines 234 and 235, they mention Fig S4A and S4B, but the figure is given in the appendix as Fig4A, panels A and B. This is confusing. Appendix A should become supplementary material, as a separate file and figures within called Figure S1, S2 ecc.

7.       Figure 4: the authors have measured binding of the 3 most studied nanobodies to hemagglutinin. In general, it would have been useful to have a measurement of binding affinity of these 3 antibodies to the 3 different recombinant target molecules and report these in a table format.

Reviewer 2 Report

I found this article refreshing for advancing antibody therapeutics. Their design, originally published twice before this manuscript, is still very innovative and the finding in this report informative to the overall antibody field. 

Line 63. "have" to and not "has" to.

I noticed two other grammatical errors. 
